# Implications of Gut Microbiota in Complex Human Diseases

**DOI:** 10.3390/ijms222312661

**Published:** 2021-11-23

**Authors:** Dahai Yu, Xin Meng, Willem M. de Vos, Hao Wu, Xuexun Fang, Amit K. Maiti

**Affiliations:** 1Key Laboratory for Molecular Enzymology and Engineering of Ministry of Education, School of Life Sciences, Jilin University, 2699 Qianjin Street, Changchun 130012, China; mengxin17@mails.jlu.edu.cn (X.M.); fangxx@jlu.edu.cn (X.F.); 2Laboratory of Microbiology, Wageningen University, Dreijenplein 10, 6703 HB Wageningen, The Netherlands; willem.devos@wur.nl; 3Human Microbiome Research Program, Faculty of Medicine, University of Helsinki, 00014 Helsinki, Finland; 4Vascular Biology Program, Department of Surgery, Boston Children’s Hospital and Harvard Medical School, Boston, MA 02115, USA; hao.wu3@childrens.harvard.edu; 5Department of Genetics and Genomics, Mydnavar, 2645 Somerset Boulevard, Troy, MI 48084, USA

**Keywords:** gut, microbiota, disease, bacteria, FMT

## Abstract

Humans, throughout the life cycle, from birth to death, are accompanied by the presence of gut microbes. Environmental factors, lifestyle, age and other factors can affect the balance of intestinal microbiota and their impact on human health. A large amount of data show that dietary, prebiotics, antibiotics can regulate various diseases through gut microbes. In this review, we focus on the role of gut microbes in the development of metabolic, gastrointestinal, neurological, immune diseases and, cancer. We also discuss the interaction between gut microbes and the host with respect to their beneficial and harmful effects, including their metabolites, microbial enzymes, small molecules and inflammatory molecules. More specifically, we evaluate the potential ability of gut microbes to cure diseases through Fecal Microbial Transplantation (FMT), which is expected to become a new type of clinical strategy for the treatment of various diseases.

## 1. Introduction


**Highlights**


1. In dysbacteriosis, the diversity and richness of bacterial populations are greatly reduced. It induces inflammation and metabolic dysfunctions, which are associated with complex diseases (the disease caused by multiple factors), such as, metabolic, gastrointestinal and neurological diseases and digestive tract cancers.

2. Bacteria can also affect health through metabolic products, such as Short Chain Fatty Acids (SCFAs), microbial enzymes, toxic metabolites and other fatty acids, etc.

3. Probiotic bacteria in the intestinal microbiota play a beneficial role for human health and improve wellness for a variety of diseases. Fecal Microbial Transplantation (FMT) and dietary intervention have positive outcome to diseases.

A large number of microorganisms are implanted in the body at birth, and their presence is detected in the skin, gastrointestinal tract, genitourinary tract and the oral cavity [1,2]. Among healthy adults, large intestinal microbiota includes Firmicutes, Bacteroidetes, Actinomycetes, Proteobacteria and Verrucomicrobia, which possess the largest densities with strongest metabolic activities [3]. Many factors, such as pH, oxygen concentration, host secretions, nutrient availability and immune defense affect microbial colonization [4].

Intestinal microbiota plays an active role in the host. The human body, as a host, provides nutrients for gut microbes, which can break down hard-to-digest carbohydrates and fatty acids, thus producing Short Chain Fatty Acids (SCFAs) that are beneficial to the human body. Each individual produces different amounts and sizes of SCFAs that regulate the many physiological pathways and the body’s immune responses [5,6]. SCFAs include butyrate, propionate and acetate. The first two have regulatory effects on intestinal physiology, immune function and a protective effect in the colon. The acetate is the substrate for lipogenesis and gluconeogenesis [7]. In addition, gut microbes help transmit hormonal signals and produce vitamins. Biofilms that are produced at the site of host-microbe interaction protect against host immune attacks by pathogenic bacteria [8].

It is well established that not all types of gut microbiota are beneficial for host survival. Gut bacteria such as *Escherichia coli*, *Fusobacterium nucleatum* and *Helicobacter pylori* can cause pathogenic reactions to the host [9]. Secondary fatty acids and toxic metabolites can cause intestinal damage in different ways. Compounds such as phenol, ammonia, para-cresol, hydrogen sulfide and amines induce inflammation, DNA damage and intestinal leakage to cause cancer [10]. These processes could be inhibited by dietary fibers or plant-based food consumption, suggesting that intestinal microbiota also play a critical role in the fermentation of carbohydrates [11].

When the dietary residues enter the colon from the small intestine, complex carbohydrates, protein residues and primary bile acids affect its composition through functions of the gut microbiome to keep the colon healthy through fermentation. In a balanced diet, the fermentation of carbohydrates plays a major role in the production of SCFAs. Conversely, in an unbalanced diet, both protein fermentation and the toxification of bile acids increase, which can promote inflammation and damage to colon cells, leading to colon cancer [12]. In addition, changes in diet affect the distribution of intestinal microbiota. Individuals who consumed more fiber had higher *Prevotella* spp. while individuals with more proteins and fats in their food had higher *Bacteroides* spp. [13]. These findings indicate that *Prevotella* spp. has better fiber degradation ability compared to *Bacteroides.* The Low Gene Count (LGC) bacterial community mainly consists of *Bacteroides* spp., which are involved in controlling obesity and metabolic syndrome [14]. When obese patients with LGC were placed under a control diet, the diversity of the microbiome increased significantly. As a result, it can be deduced that diet affects the composition of the gut microbiome and modifies the host’s absorption of nutrients and changes the host’s ability to develop an appropriate immune response [15]. Human intestinal microbiota contains a large number of Antibiotic Resistance Genes (ARGs) [16] and tetracycline resistance genes account for a large proportion among them. These ARGs can be freely transferred between other intestinal bacteria residing within the human colon [17]. The use of antibiotics may help to enrich ARGs in human intestinal microbiota and a longer usage of antibiotics increases the chance of harboring more ARGs [18,19]. Through DNA microarray analysis, it has been shown that the ARGs of human intestinal microbiota can be accumulated in adulthood, which create more complex problems with age [20].

The gastrointestinal tract is composed of a neural network containing 200 to 600 million neurons, which form the Enteric Nervous System (ENS) and control gastrointestinal physiology as reviewed by Furness, 2012 [21]. Intestinal microorganisms can regulate the central nervous system through immune, circulation, and ENS [22,23]. However, once the microbial composition is imbalanced, obesity, diabetes, gastrointestinal diseases, neurological diseases and allergic reactions can be induced [24,25].

With respect to treatment, a balanced diet with Fetal Microbial Transplantation (FMT) and oral administration are used. The positive outcome of these treatments provides a theoretical basis for new ideas of targeting intestinal microbiota in the future. This review will focus on three aspects: intestinal microbiota and diseases, the mechanism of intestinal microbiota on human diseases and, the therapeutic potential of intestinal microbiota.

### Methods of the Review

This review aims to develop a general concept of the impact of microbiota on human health and complex diseases. After planning the outline of the subject, each of the topic of subheadings are subjected to a search to obtain relevant literature on “PUBMED”. The initial search term was “gut microbiota microbiome”, as some authors used the term “microbiota” or “microbiome”. Then, with this initial search term each disease name was added in a further search. A PRISMA (Preferred Reporting Items for Syestematic review and Meta Analysis) flow diagram is shown in Figure 1 [26]. As the role of microbiota in human diseases are extensive and the literature mainly spanned from 1980 to 2021, we emphasized the relevance of the time period of 2010–2020 as providing the most important research advancements in this area. Bacterial nomenclature and taxonomy were updated based on the website https://www.ncbi.nlm.nih.gov/Taxonomy/Browser/wwwtax.cgi (accessed on 11 October 2021). Lactobacillus nomenclature was updated based on https://www.ncbi.nlm.nih.gov/Taxonomy/Browser/wwwtax.cgi (accessed on 11 October 2021).

## 2. Gut Bacteria and Human Health

Human lifestyle influences the composition of intestinal microbiota. In turn, intestinal microbiota also affects human health, including functions such as the metabolism, intestinal function, and the neurological and immune system. A flow chart depicting the role of intestinal bacteria and a list of bacteria that are associated with various diseases are shown in Figure 2 and Table 1 respectively.

### 2.1. Digestive Tract Diseases

The gastrointestinal tract (GIT) is a complex combination of microorganisms. Mammalian GIT contains a variety of microbial communities, including viruses, bacteria and, fungi [27]. Changes in the functional composition of intestinal microbiota can cause metabolic disorders, particularly inflammatory bowel disease (IBD) or Chron’s disease.

In normal conditions, the small intestine produces a specific antimicrobial immunoglobulin (IgA) against disease-causing bacteria. Secretory IgA (SIgA) plays a critical role in the establishment and maintenance of gut bacteria. Additionally, some IgAs promote bacterial growth or do not have an impact on bacterial survival [28]. However, cross-specificity or poly-specificity is not uncommon [29]. IgA-SEQ using 16S ribosome identified four groups of intestinal microbiota, *Bacteroides*, *Lactobacillus*, UC *Erysipelotrichaceae*, and Segmented Filamentous Bacteria (SFB) that induce IgA in the host [30]. Using a specialized IgA binding technique, colitis causing bacteria, Prevotellaceae, *Helicobacter* spp., Flexispira and SFB were also identified in the intestinal tract of the mouse model. The symbiotic microbiota inhibits the entry of pathogens into the intestinal epithelial barrier by scavenging for nutrients, thus maintaining the balance of the epithelial barrier and the host immune response. *Firmicutes* and *Bacteroides* are mainly involved in the secretion of mucus, SCFAs and the activation of certain pathways in the immune system [31]. In patients with IBD, the abundance of intestinal symbiotic microbiota decreases, leading to a decreased secretion of antimicrobial peptides (α-defensins) from Paneth and Goblet cells. In fact, IBD patients show a reduction of *firmicutes* and an increase of *Proteobacteria* [32]. The increased *Proteobacteria* also leads to the reduction of SCFAs that induce excessive production of IgG to target the symbiotic microbiota. Activated macrophages produce pro-inflammatory cytokines that overstimulate Th1 or Th17 cells to induce inflammation [31]. The loss of intestinal barrier integrity leads to increased bacterial antigen translocation and the stimulation of the inflammatory response of intestinal mucosa, which is the core pathological feature of IBD. Intestinal mucus and cells’ tight junctions play a major role in maintaining the integrity of the intestinal barrier. Mice lacking claudin 7, Hnf4a and Muc2 all spontaneously contract to colitis [33]. In muc2−/− mouse, the increased permeability of the epithelial cell barrier involving tight junctions also show mucus deficiency, which is accompanied by claudin gene expression disorder [34]. This codependence may be caused by the signal imbalance of the regulation of mucus and tight junctions. This interdependence contributes to the feedback mechanism in the inflammatory response, making the IBD permanent and causing it to perpetuate. *Salmonella* infection on IBD patients forms membrane folds or interrupts the tight junctions in epithelial cells, thereby inducing inflammation in patients. Once it enters epithelial cells, it blocks the autophagosome pathway, preventing self-degradation [35]. *Salmonellae typhimurium* enter the neutrophils in the intestinal lumen, where ROS (Reactive Oxygen Species) and tetracycline are produced. These agents act as the electron acceptor of the *Salmonella* electron transport chain, thus making abundant growth of *S. typhimurium*, accounting for a major microbial population. Abundant *Salmonella* population reaches the Peyer’s patches by invading dendritic cells (DCs) in the epithelial barrier or M cells in the lumen and can be identified by the presence of their glycoprotein-2 [31].

**Table 1 ijms-22-12661-t001:** Microbiota associated in diseases. The abundance of bacteria identified in various diseases.

Cells/Tissue/Other	Host	Disease	Methods	Bacteria Identified/Increased	References
Small intestine	Human	Digestive tract diseases	16S RNA	*Bacteroides*, *Lactobacillus*, UC *Erysipelotrichaceae*	[28]
Intestinal tract cells	Mouse	Digestive tract diseases	IgA seq	Prevotellaceae, *Helicobacter* spp., Flexispira and SFB	[30].
Ilium and rectum Biopsy	Human	IBD	16S RNA	*Proteobacteria*	[32]
ileocecal biopsies	Human	PSC-IBD and UC	16S RNA	*Escherichia, Lachnospiraceae* family, *Veillonella* and *Megasphaera*	[36]
Large intestine	Human	IBS	RT-QPCR	*Lactobacillus*, bifidobacteria, and *Clostridium*	[37]
Colon and caecum	Mice	Osmotic diarrhea	16S RNA	*Bacteroides*	[38].
Blood, stool, urine	Human	Anxiety and Depression in IBS	fMRI	*Bifidobacterium longum* NCC3001	[39]
Fecal sample	Human	ASD	16 S RNA	*Faecalibacteriu, Ruminococcus,**Sarcina* and *Clostridium*	[40]
Fecal sample	Human	Schizophrenia	Magnetic Resonance Spectroscopy	*Clostridium, Lactobacillus* and *Bacteroides*	[41]
Frozen brain Biopsy	Human	Alzheimer’s	16S RNA and Nextgen Sequencing	*P. gingivalis, F. nucleatum* and *P. intermedia, Helicobacter pylori*	[42]
Fecal sample	Human	Parkinson Disease	RT-QPCR	Enterobacteriaceae	[43]
Fecal sample	Human	Diabetes	16S RNA	Bacteriodes, bifidobacteria, Clostridium	[44]
Cecum	Mice	Obesity	16S RNA	Firmicutes and Bacteriodetes	[45]
Fecal sample	Human	Gaut	16S RNA	*Bacteroides, Porphyromonadaceae Rhodococcus*, *Erysipelatoclostridium* and *Anaerolineaceae*	[46]
Fecal sample	Human	Hypertension	Metagenome shotgun sequencing	*Klebsiella* spp., *Streptococcus* spp., and *Parabacteroides merdae*	[47]
Fecal sample	Human	Atherosclerosis	Metagenome shotgun sequencing	*Roseburia*	[48].
Fecal sample	Human	Colorectal cancer	RT-QPCR	*F. nucleatum,* *Peptostreptococcus anaerobius* *, P. stomatis, Solobacterium moorei, Gemella morbillorum and Parvimonas micra*	[49]
Fecal sample	Human	Colorectal Cancer	16S RNA, NextGen sequencing	*F. nucleatum*	[50].
Fecal sample	Human	Allergy	16S RNA	Enterobacteriaceae and Parabacteroides	[51].

Primary Sclerosing Cholangitis-IBD (PSC-IBD) is a hepatic-biliary-intestinal axis-associated inflammatory autoimmune disease. *Escherichia, Lachnospiraceae* family, *Veillonella* and *Megasphaera* were significantly elevated in patients with PSC-IBD with a significant increase in adhesion compared to controls. The genes of these bacteria encode an amine oxidase that acts as a vap1 substrate [36]. VAP1 recruits effector cells to the liver, both as an adhesion molecule and as a semi-carbazide-sensitive amine oxidase. In absence of VAP1, *Prevotella* and *Roseburia* spp. (butyrate producers) in IBD-PSC patients are reduced, with the complete removal of *Bacteroides* spp.

Gwee et al [52] were the first to demonstrate that patients with high stress and mental disorders had a higher risk of contracting IBS (Irritable Bowel Syndrome). In IBS patients, *E. coli* and *Salmonella* were found to cross through the epithelium and to increases vasoactive intestinal peptide (VIP) levels, tryptase levels, mast cell counts, and mast cells expressing VIP receptor type 1 (VPAC1) in plasma. Significant changes in gene expression were observed in IBS-D patients carrying *Lactobacillus*, bifidobacteria and *Clostridium.* Probiotics improved the health of these patients [37]. The IBS-C patients do not have these three bacteria, but the presence of a high amount of methane indicates that the pathogenesis of IBS-C patients may also be related to methanogen bacteria.

Dysbiosis, or the imbalance of microbiota, especially in the gut, is shown to be associated with the onset and progression of IBS [53]. Functional dyspepsia (FD) is a frequent gastrointestinal disorder characterized by epigastric pain or burning of the upper digestive tract and is associated with differences in the commensal bacterial community between FD patients and healthy controls [54]. Intestinal dysbiosis is an emerging concept and its evaluation in IBS and FD could dictate future treatment strategy [55]. *Helicobactor pylori* associated dyspepsia is considered as a distinct entity of gastrointestinal disorder and could be treated by eradication therapy [56].

Osmotic diarrhea, a disease of intestinal osmotic pressure occurs in patients with celiac disease, malabsorption, lactose intolerance and the abuse of laxatives. Polyethylene glycol (PEG) exposes mice to mild diarrhea. After 6 days of dose dependent PEG treatment, a significant reduction of the microbial diversity was observed, S24-7 bacteria and Y-Proteobacteria were disappeared, and the number of low-abundance microbiota, such as *Bacteroides* were amplified. However, after stopping the treatment, the physiological damage of the host’s intestines was recovered, and colonization of the bacteria in the intestines was re-established [38].

Lyer et al. [57] found that dietary and microbial oxazole can induce inflammation by modulating the aryl hydrocarbon receptor (AhR) response. Oxazole compound OxC, derived from industrial, dietary, and intestinal commensal bacteria can impair the lipid antigen presentation function of CD1d in intestinal epithelial cells, reduce IL-10 production, promote iNKT-mediated colon inflammation and can activate the AhR that triggers a CD1d-dependent intestinal inflammatory response.

### 2.2. Intestinal Encephalopathy

The intestinal microbiota plays an irreplaceable role in the basic processes of neurogenesis, such as the formation of the blood-brain barrier, the development of the myelin sheath, the maturation of microglia and brain development [58,59,60]. Sterile mice (GF, germ free) without intact intestinal microbiota showed more ADHD (Attention Deficit Hypertension Disorders) symptoms and risk-taking behavior than traditional (non-specific pathogen-free) mice. They also demonstrated deficiencies in learning and memory skills [61,62]. In addition, the Blood-Brain Barrier (BBB) of GF mice is impaired, which is accompanied by an increase in prefrontal myelination, serotonin receptors (5-HT1A), neurotrophic factors (such as BDNF) and NMDA receptors in the hippocampus [63]. Animal models indicate that intestinal microbiota affects neuropsychiatric disorders, including depression and anxieties in autism (ASD) [64], schizophrenia [65], Parkinson’s disease (PD) and Alzheimer’s disease (AD) [66].

#### 2.2.1. Anxiety and Depression

Depression is associated with stress that could be accompanied by a decrease in the diversity or density of the intestinal microbiota. Intestinal bacteria produces LPS (Lipopolysaccharides) that can enter the bloodstream to cause inflammation [67]. On the other hand, excessive saturated fatty acids, produced by gut bacteria, can induce pro-inflammatory factors, which are produced by adipocytes and macrophages and disrupt the integrity of the BBB [Figure 3]. These pro-inflammatory immune cells reach the brain, causing neuroinflammation that affects mood (depression) and behavior [68,69]. In comparison to healthy people, patients with major depression have an increased abundance of thick-walled bacteria, actinomycetes and *Bacteroides*. When sterile mice receive the intestinal microbiota of patients with severe depression, they exhibit depression-like behavior compared to mice that receive healthy human intestinal microbiota [70]. *Bifidobacterium longum* NCC3001 can increase the quality of life of patients with IBS and reduce the response to multiple negative emotional stimuli in the brain of patients, which has a regulatory effect on depression [39].

#### 2.2.2. Autism Spectrum Disorders (ASD)

Patients with autism often exhibit gastrointestinal dysfunction. C57 mice of an ASD model display abnormalities in behavior, immune state and an intestinal ecological imbalance [71]. These differences are associated with abundances in *Bacteroides*, *Parabacteroides*, *Sutterella,*
*Dehalobacterium* and *Oscillospira* [72]. Similarly, the abundance and diversity of *Clostridium* spp., anaerobic and micro-aerophilic bacteria without spore formation also increases in ASD patients compared to control groups [40]. The ASD gut microbiota lacks specific bacteria such as *Prevotella copri* [73,74]. Gastrointestinal complications are more common in children with ASD than controls and *Sutterella* is closely associated with intestinal epithelial cells in children with ASD [75]. Therefore, enriching the microbiota with specific microorganisms may contribute to the treatment of ASD [76].

#### 2.2.3. Schizophrenia

Socially Isolated (SI) rats are useful animal models for studying schizophrenia. In the intestinal microbiota of SI rats, *Actinomycetes* are increased and, *Clostridium* is decreased. In addition, early life stress causes long-term changes in the intestinal microbiota, causing abnormal neurodevelopment and behavior. By analyzing the brain with magnetic resonance spectroscopy in patients with ultra-high risk (UHR) of schizophrenia, we observed that *Clostridium, Lactobacillus* and *Bacteroides* in the intestinal microbiota of UHR group were significantly higher than for the HR (high risk) and HC (healthy control) groups [41]. Changes in intestinal microbiota led to differences in the choline concentration in the Anterior Cingulate Gyrus (ACG) and, the choline levels in the UHR group were significantly increased in comparison to the HR and HC groups. Elevated choline levels are hallmarks of membrane dysfunction in brain images of Schizophrenia patients.

In schizophrenia patients, *Chlamydia* infections increase the diversity of blood microbiota, which negatively correlates with the abundance of CD8^+^ memory T cells [77]. The diversity of the microbiota significantly increased in the blood of patients with schizophrenia with two special microbes, Planctomycetes and *Thermotogae* [78]. Planctomycetes are Gram-negative phylum of bacteria, which forms a Planctomycetes-*S. cerevisiae* association with *Chlamydomonas*.

There is also a difference in fecal microbiota between schizophrenia patients and healthy people [79]. A significant increase in six types of bacteria (*Succinivibrionaceae, Collinsella, Megasphaera, Clostridium, Methanobrevibacter* and *Klebsiella)* and a decrease in three types of bacteria *(Blautia, Roseburia* and *Coprococcus)* were documented.

#### 2.2.4. Alzheimer’s Disease (AD)

The diversity of intestinal microbiota is reduced in AD patients compared to healthy controls. Thick-walled bacteria and *Bacteroidetes* are abundant while actinomycetes and bifidobacteria are reduced. Moreover, the difference in types of bacterial genus between patients and the healthy controls are correlated with Cerebro Spinal Fluid (CSF) biomarkers of an AD pathology [80]. There are several types of CSF biomarkers: (1) Aβ42/Aβ40: Lower Aβ42/Aβ40 in CSF reflects lower amyloid content, which means increased amyloid deposition in the brain. (2) The p-tau: a marker of neurofibrillary tangles, the higher the p-tau is, the more severe is the tangles in the brain [81]. (3) p-tau/Aβ42: a higher p-tau/Aβ42 indicates higher pathologic complications of AD. (4) Chitinase-3-like protein 1 (YKL-40): an increase in YKL-40 (astrocyte and microglial activation markers) in the CSF of patients denotes dementia caused by AD [82]. An increase of the *Bacteroides* and *Blautia* in AD, is negatively correlated with CSF Aβ42/Aβ40, and positively correlated with CSF p-tau and p-tau/Aβ42. Furthermore, the increase in *Bacteroide*s and the decrease in *Turicibacter* and SMB53 (family Clostridiaceae) in AD patients is associated with an increase in CSF YKL-40.

*H. pylori* infection influences the pathophysiology of AD [83]. Although it induces severe gastritis and neuroinflammation, it does not induce amyloid deposition or systemic inflammation [84]. The intestinal microbiota of AD patient may be dysregulated, including *Escherichia* and *Shigella,* which are increased, whereas the anti-inflammatory effect of *E. coli* and *Bacteroides fragilis* decreases [85]. These changes lead to elevated levels of related inflammatory factors in the blood and brain, such as IL-6, IL-1β, IL-17A, IL-22, ROS, NFkB, CD14, TLR1/2, which trigger neurodegeneration. A decrease of the abundance of *Lactobacillus* and bifidobacteria in AD patients, results in the decreased expression of GABA in blood and brain with a cognitive impairment. When probiotics of these two bacteria are supplemented, the cognition of AD patients is improved [42,86]. Therefore, poor conditions of AD patients may be associated with altered intestinal microbiota, which can be used to treat AD.

LPS, produced by bacteria, has also been found to be highly enriched in the neocortex and hippocampus of AD patients, exhibiting a 2-fold and a 3-fold increase, respectively, compared to healthy controls. LPS enrichment is more severe in some patients with advanced AD and can be increased by a factor of 26 [87]. In addition, environmental factors can also indirectly aggravate AD. Chronic noise causes cognitive impairment in young SAMP8 mice and beta amyloid (Aβ) enrichment in the hippocampus may be associated with decreased intestinal microbiota diversity [88]. In younger SAMP8 mice, endothelial tight junction proteins are reduced in the gut and brain, while serum neurotransmitters and inflammatory mediators are elevated. Noise-induced intestinal bacterial changes are further confirmed by bacterial transplantation experiments, destroying epithelial integrity and enriching Aβ.

#### 2.2.5. Parkinson’s Disease (PD)

Microbial density on the mucosal surface in the intestinal and olfactory bulbs differs in PD patients and healthy controls, suggesting that there could be an involvement of microbes in PD pathogenesis [89]. In the fecal samples of PD patients, the abundance of *Prevotella* and *Bacteroides* are decreased and that of the family *Enterobacteriaceae* is increased. At the same time, the concentration of SCFAs decreases significantly, which may lead to changes in the CNS and may indirectly cause gastrointestinal motility disorder in PD patients [43]. The risk of PD may be significantly higher in patients infected with *H. pylori* than in healthy people over 60 years old, but not in people under 60 years old. However, no significant association between *Helicobactor* eradication and PD risk are observed [90]. Rotenone is a common inhibitor of mitochondrial complex I and in PD models, microbiota may promote the rotenone induced-PD via motor and central nervous system dysfunctions [91].

### 2.3. Metabolic Diseases

Increased intestinal permeability and increased bacterial antigen in blood can be used as indicators for the validation of metabolic syndrome.

#### 2.3.1. Diabetes & Obesity

The development of diabetes is shown to be accompanied by changes in the intestinal microbiota. Compared to healthy controls, Gestetional Diabetes Mellitus (GDM) patients have considerable differences in the intestinal microbiota [92]. Most of the bacteria of phylum Actinobacteria and of the genus *Collinsella, Rothia and Desulfovibrio* are abundant in these patients. Furthermore, the microbiota composition of GDM patients is abnormal even after 8 months of delivery. This study also suggests that *Akk**ermansia muciniphila* is associated with lower insulin resistance, and *Christensenella* is associated with higher fasting blood glucose levels. When metabolic syndrome (MS) is induced by a high fat diet (HFD) the composition of the intestinal microbiota may be modified [93]. Supplements of *Lactobacillus paracasei* CNCM I-4270 (LC), *L. rhamnosus* I-3690 (LR) and *Bifidobacterium animalis* subsp., could transform the whole composition of intestinal microbiota, which is destroyed by HFD of lean mice. *Lactobacillus* and *Bifidobacterium* reduce obesity to a certain extent by influencing the intestinal microbiota composition in mice.

An analysis of the fecal bacterial metabolites in children at risk for T1D revealed that they had more fecal pro-inflammatory compounds than normal infants [44]. The diversity of intestinal microbiota in diabetic children decreases before the onset of the disease [94]. A 16s ribosomal high-throughput sequencing confirms an imbalance and reduction of diversity of *A. muciniphila* in the feces leads to the development of autoimmunity in T1D compared to non-diabetic children [44,95]. T1D children have impaired intestinal permeability, higher proinflammatory cytokines and lipopolysaccharides in the serum as well as altered intestinal microbiota gene expression, with increased lipid and amino acid metabolism [96]. When the cecum microbes of adult males were transferred to young females, changes in the sex hormones and phospholipids in the serum were observed. Both the host metabolism and islet inflammation were also associated with the gut microbiome in the T1D high-risk individuals [97]. Firmicutes and *Bacteriodes* levels are increased in the intestinal microbiota of T1D patients, while they are reduced in T2D and obese patients [44,95]. When the host-microbe interacting microbiota was absent in newly onset T1D patients, the absence of host proteins that maintain mucosal barrier, microvilli adhesion and pancreatic exocrine function in the serum was observed [98]. The imbalance of intestinal microbiota may promote an inflammatory environment in the intestinal tract, thereby promoting T1D, and affecting the inflammatory mediators in host tissues.

The proportion of Firmicutes and *Bacteroides* are found to be increased compared to their lean peers (ob/+ or +/+) mice, and changes in the microbiome are associated with an increased ability to obtain energy from the diet [45]. When the ob/ob mice microbiome was transferred to aseptic mice, an increase in body fat was detected although food intake was constant throughout the process [99]. Similar changes in microbiome composition were also observed in diet-induced obese mice (DIO), mainly when Mollicutes were transferred, which promote the processing of monosaccharides [100]. The transfer of the intestinal microbiome from DIO mice increases the weight of the individual mice. It may be accompanied with increased intestinal inflammation with bacterial lipopolysaccharides and enhanced microbial DNA translocation [101]. An intake of *A**. muciniphila* by mouth increased the species diversity of intestinal microbiota in diabetic rats [102]. These bacteria also improve the liver function, reduce sugar and fat toxicity, relieve oxidative stress, inhibit inflammation, restore normal intestinal microbiota, and ease T2D symptoms. Non-obese diabetes mellitus (NOD) mice had an insufficient expression of β-defensin 14 (MBD14) in their pancreatic endocrine cells and treatment with MBD14 significantly reduced the development of mice’s autoimmune symptoms [103]. MBD14 promotes the secretion of IL-4 by specific B cells through TLR2, thereby improving the balance of M1-M2 macrophages and inducing regulatory T cells. Intestinal microbiota produces AhR ligands and butyric acid to promote the secretion of IL-23 and IL-22 from the pancreatic islet lymphocytes (ILC) that induce pancreatic endocrine cells to express MBD14, thereby preventing autoimmune diabetes. It is also observed that the use of antibiotics in early life can permanently alter the intestinal microbiota of NOD mice and leads to changes in metabolites in the cecum, liver and serum. With the increase of antibiotic treatment, a significant reduction in the microbiota diversity accelerates T1D by affecting innate and adaptive immune responses [104].

Imidazole propionate (ImP), an histidine derived metabolite produced by intestinal microbiota is higher in blood of patients with T2D [105]. In an in vitro simulated fermentation experiment, T2D patients displayed higher ImP levels, produced by the fermentation of fecal bacteria, than the healthy controls. ImP injection reduces the insulin receptor (IRS) phosphorylation in mice, leading to the disruption of sugar tolerance levels. In women with HIV infection, diabetes is significantly related to the composition of intestinal microbiota and plasma metabolites, especially tryptophan metabolism byproducts [106]. The abundance of four bacterial genera including *Leptotrichia* and *Trevisan* was relatively low in the diabetic female patients. The tryptophan metabolites in their plasma were relatively high, while the glycerophospholipids were relatively low.

#### 2.3.2. Other Diseases with Abnormalities in Metabolism

Other diseases such as gout, arthritis, cardiovascular and cerebrovascular diseases, although characterized by abnormalities in the metabolism, are not classified as metabolic diseases. These diseases have also been confirmed to be associated with intestinal microbiota. Compared to healthy controls, patients with gout or HUA (Hyperuricemia) had significantly higher levels of *Prevotella intermedia* in their saliva, but significantly lower levels of *Serratia marcescens* [107]. The composition of intestinal microbiota in gout patients was deficient with *Faecalibacterium*
*prausnitzii* and rich with *Bacteroides caccae* and *B. xylanisolvens* [108]. In the fecal microbiota of gout patients, the number of opportunistic pathogens such as *Bacteroides*, *Porphyromonadaceae*, *Rhodococcus*, *Erysipelatoclostridium* and *Anaerolineaceae* was increased [46].

The collagen-induced arthritis (CIA) model showed significant intestinal dysbacteriosis and mucosal inflammation before the onset of visible arthritis which were further aggravated during the course of the diseases [109]. Before the onset of arthritis, the *Lactobacillus* was enriched in CIA-sensitive mice [110]. With the development of the disease, the abundance of *Bacteroidaceae* and *Lachnospiraceae* increased significantly. In rheumatoid arthritis (RA), several bacterial species were found to be abundant in the oral cavity [111]. Compared to healthy people, fecal microbiota of patients with spinal arthritis and RA show disease specificity and reduced diversity of microbiota [112]. The intestinal microbiota of Italian adolescent patients with idiopathic arthritis was lower than that of healthy controls [113]. It was found that *Allobaculum* was decreased, and that *Erysipelotrichaceae* and *F. prausnitzii* were increased.

##### Common Cardiovascular Diseases including Hypertension, Atherosclerosis and Chronic Hypertension

Recent studies have shown a direct link between intestinal microbiota and blood pressure control in animal models [114]. The composition of the microbiota of African origin and Caucasian patients with hypertension may be different, resulting in significant differences in their function, the synthesis of amino acids and inflammatory antigens [115]. Angotensin II-injected sterile mice showed angiotensin II-induced vascular dysfunction and hypertension [116]. The blood pressure reduces in drug-resistant hypertension patients, once they are treated with antibiotics and a combination of microbiota [117]. In overweight and obese pregnant women, it has been found that the higher the bacterial butyrate production by genus *Odoribacter*, the lower the blood pressure [118]. Many hypertensive patients with excessive bacterial species are associated with intestinal inflammation and intestinal barrier dysfunction [119]. The number of intestinal bacteria and fungi in faeces of chronic heart failure patients increased with increased intestinal permeability compared to the healthy control group [120].

Yan et al (2017) identified numerous differentially expressed microbial genes in hypertensive patients [47]. Around 69% of these genes are clustered into 68 Macrogenome Linkage Groups (MLGS), which can describe microbial differences at the species level. In hypertension patients, *Klebsiella variicola* and *Streptococcus* MLGs are higher whereas in the control group, *Roseburia* and *F. prausnitzii* MLGs are higher. Besides, the enriched MLGs in hypertensive patients consists of several *Bacteroides* spp. including *B. eggerthii*, *B. cellulosilyticus*, *Pyramidobacter piscolens* and *Sutterella wadsworthensis*. The MLGs that are enriched in controls include several other *Bacteroides* spp. including *B. uniformis*, *B. nordii*, *B. dorei*, *Aeromicrobium massiliense* and *Megasphaera micronuciformis*. Four other species, namely, *Prevotella 2*, *Prevotella 7*, *Tyzzerella*, and *Tyzzerella 4* are abundant in patients with high cardiovascular risks whereas two species of bacteria *Alloprevotella* and *Catenibacterium* do not exist in patients with high cardiovascular risk [121].

The same bacterial groups observed in atherosclerotic plaques also exist in the intestinal tract, which may impair the stability of the plaque [122]. Metagenomic sequencing of fecal microbiota showed the differences in microbial composition between unstable and stable plaques. The unstable plaque was associated with a decrease in the genus *Roseburia* in feces, with both enhanced proinflammatory peptidoglycans and anti-inflammatory carotenes [48].

### 2.4. Cancer

Gut microbes can increase the incidence of certain diseases, such as cancer. Some bacteria promote the development of cancer through different mechanisms. For example, (1) Genotoxic polyketide synthase (PKS)-harboring *E. coli* can directly damage DNA [123]; (2) *F. nucleatum* mediated E-cadherin/beta-catenin signaling through FadA adhesin increases cancer probability [124]; (3) *P. anaerobius* induces cholesterol biosynthesis and cell proliferation through TLR2 and TLR4 pathways, leading to cancer signaling [125].

Colorectal cancer (CRC) patients exhibit an abundance of *F. nucleatum, Peptostreptococcus anaerobius**, P. stomatis, Solobacterium moorei, Gemella morbillorum and Parvimonas micra* and which together to form a network of bacteria [49]. When normal and germ-free mice were fed with CRC patients’ or healthy feces, CRC mice developed highly atypical hyperplasia and showed a significant increase in the proportion of polyps. They also exhibited germ cell proliferation, reduced fecal bacteria abundance and an altered microbiota composition [126].

CRC microbiome can also induce multiple inflammation and activate carcinogenic pathways. The expression of cytokines (CXcr1/2, IL-17a, IL-22, and IL-23a), responsible for regulating inflammation in the small intestine of mice was increased. They also presented an increased incidence of tumors in the colon with an increase in the proportion of Th1 and Th17 cells. Therefore, the microbial product may not act only as a key component in CRC, but also induces the development of CRC through the activation of Th17 pathway [127,128].

Oral microbiota also plays a role in CRC development. The common form of *Fusobacterium nucleatum (Fn)* in the oral cavity promotes colon tumor in animal models, as is evidenced by the enrichment of *Fn* in CRC intestine [50]. *Fn* may be responsible for a possible link between the oral cavity and intestinal mucosa, which may further lead to the maladjustment of intestinal microbiota, the destruction of intestinal homeostasis and changes in the microenvironment, leading to the development of CRC [129]. In addition, *Fn* can be detected in liver metastasis and with paired primary metastatic CRC [130], suggesting that *Fn* may be transplanted through either the hematological or lymphatic pathways. *Fn* was primarily found in the proximal colon cancer and the proportion of *Fn* in CRC gradually increased from the rectum to the cecum [131].

Recently, *Fn*’s two outer membrane proteins, Fap2 and FadA, have attracted attention. The Fap2 protein directly interacts with TIGIT, which is expressed on NK cells and tumor infiltrated lymphocytes, thereby inhibiting the cytotoxicity and lymphocyte activity of NK cells [132]. A recent study reported that *Clostridium* protein-dependent host polysaccharides were enriched in tumor sites and Fap2 protein was specifically bound with overexpressed polysaccharide glycan GalGalNAc (d-galactate-chuan-(1-3) -acetyl-d-galactosamine) on the surface of CRC cells. This binding of *Fn with* Fap2-GalGalNAc was reduced on the surface of CRC cells due to less Fn. When the *Fn* was replenished by injection, the bacteria relied on Fap2 to locate tumor tissues in mice indicating that it reached the colorectal adenocarcinoma through blood. Fap2 also prevents the host immune system from specifically killing tumor cells [133]. FadA protein plays an important role in cell attachment. FadA binding to e-cadherin can further activate the beta-catenin signal, resulting in an increased expression of oncogenes, WNT and inflammatory genes [124]. However, another study showed that the presence of FadA and Fap2 adhesins in *Fn* were not sufficient to induce inflammation or cancer in a preclinical model of CRC mice [134]. TLR4 is an important receptor of lipopolysaccharide (LPS), which is over-expressed in CRC and may promote tumor formation [135]. CRC cells infected with *Fn* can increase the expression of miR-21 in TLR4 activated mice [136], which can act as a key inhibitor in synergistic CRC. A TLR4/P-PAK1 cascade can activate the beta-catenin signaling in CRC [137]. In conclusion, the WNT/-catenin signaling pathway seems to play an important role in the occurrence of CRC mediated by *Fn*.

CRC may be also associated with biofilm-related serious intestinal diseases. Biofilms containing enterotoxigenic *B. fragilis* and *F. nucleatum* were detected in tumor tissues [138,139]. Therefore, mature biofilms, when present in healthy tissues near CRC tissues, serve as an early warning signal for the critical transformation of the intestinal environment to imbalance, injury and pathogen infection.

### 2.5. Immune Dysfunction

There is growing evidence that the maternal microbiota during pregnancy has a significant impact on the risk of allergic diseases in offspring. The maternal microbiome can affect the immune system of the developing fetus by means of the metabolic products of the microbiome and the transmission of IgG via the placental route. Interactions between microbiome stimulating factors (such as LPS) and microbiome regulatory factors (SCFAs) may also affect fetal immune development [137].

The “hygiene hypothesis” was originally proposed for appendicitis in the early 20th century, as an immune system is not adequately trained for infections [140]. Later, in 1989, its application was extended to allergies, as allergies might be prevented by viral infections transmitted by “unhygienic contact” to siblings early in life [141]. It is shown that early childhood exposure of a particular microorganism protects against certain allergic disease by contributing to the development of the immune system [142]. From the 4th week to the 26th week after birth, the relative abundance of Paracteroidetes (*Bifidobacterium* spp.) and two enterobacterium (*Clostridium* spp. and *Lachrospiraceae* spp.) in the intestinal microbiota gradually decreases and bacteria, such as *Eubacterium* spp., *Anaerostipes* spp. which can produce butyric acid from lactic acid, gradually increases [51]. On the other hand, in the fecal microbiota of allergic children, *Ruminococcus*, *Bacteroides*, platycocci and fecal cocci decreased. At the age of 8, the fecal microbiota of allergic children is enriched with bifidobacteria, while Lactobacillus and Enterococcus are absent. In addition, *Faecalibacterium* is found to be associated with the regulation of IL-10 mRNA and FOXP3 levels [143].

Most antibiotics have an adverse effect on allergic diseases and, by destabilizing the symbiotic microbiota, they increase the risk of secondary infections, drug-resistant bacterial transmission, allergies and asthma [144]. Antibiotic use during pregnancy is associated with an increased risk of asthma in offspring [145]. The use of cephalosporin is associated with asthma, allergic rhinitis, systemic allergic reactions and allergic conjunctivitis in early childhood. The use of acid-fast drugs and antibiotics, that alter the microbiota within 6 months of birth, may contribute to allergic diseases [146]. The use of antibiotics during pregnancy changes intestinal and vaginal microbiota but may not be conducive to the colonization of postnatal microbiota. It may also be associated with an increased risk of childhood obesity [147] and early childhood asthma [148].

Evidence suggests that skin inflammation is associated with intestinal microbiota. When an anaphylactic mouse was induced by oxazolone, the intestinal microbiota increased the sensitivity of the dermatitis [149]. The colonization of *Staphylococcus aureus* was an earlier symptom of allergic dermatitis. An imbalance of intestinal microbiota increases the intestinal permeability and bacterial translocation to lymphatic organs, leading to delayed hypersensitivity. This causes allergic skin diseases in mice devoid of Mitochondrial Antiviral Signaling (MAVS) [150]. *Viral infections* involve 80% cases of acute respiratory tract illnesses (ARIs) but a small number of bacterial genus, such as *Moraxella, Streptococcus* and *Haemophilus* dominantly shift towards a nasopharyngeal microbiota (NPM) composition, causing allergies [151]. Viral infection and ARIs were associated with an increase in the abundance of disease-associated bacteria in NPM.

## 3. The Mechanism of Action of Gut Bacteria in Health and Disease Development

Most bacteria in intestine form complex networks which include beneficial bacteria and harmful bacteria. Under normal circumstances, they exist optimally and are beneficial to the health of the host, but when an imbalanced occurs, the risk of disease increases. The secretion of bacterial biofilm is also one of the factors that leads to the development of human diseases. In contrast, the metabolites of intestinal microbiota, such as SCFAs, etc, can protect the host from pathogen invasion by activating immune defense.

### 3.1. Beneficial Bacteria and Harmful Bacteria

Probiotics are a mixture of live bacteria and/or yeast that live in the body or are consumed from outside and retain the beneficial microbiota in the gut. A more precise definition was considered for “probiotics” by WHO [152] as “live microorganisms which when administered in adequate amounts confer a health benefit on the host”. Among them, probiotics in the gut microbiome alleviate many diseases and play a positive role in human health. Harmful bacteria are pathogenic.

Probiotics that are beneficial to several diseases have been identified. *Lactobacillus rhamnose* GG secretion and LSM (*Lactobacillus* Soluble Medium) regulates the function of dendritic cells (DC) by strengthening the ability of the T cell response. At this stage, IL2 and IFN-γ producing T cells and Foxp3^+^ expression were increased [153]. The expression of IL-1β, IL-12, TGFβ and ICAM1 in the colon alters the expression of miRNA and changes the ratio of *firmicutes* to *Bacteroides* [154]. In ovalbumin sensitized mice, *Lactobacillus sakei* WIKIM30 [155] reduces the level of serum IgE and IL4 and enhances TReg differentiation and IL10 in the mesenteric lymph nodes to significantly alleviate atopic dermatitis (AD) skin lesions.

*Acinetobacter iwoffii* improves respiratory hyperresponsiveness by blocking the recruitment of dendritic cells in the lungs, thereby reducing IL-13^+^ CD4^+^T cells (key to initiating early AhR) [156]. When a baby mouse was exposed to House Dust Mite (HDM), it was able to resist AhR. *Lactobacillus plantarum* P8 not only relieves stress and anxiety, but also improves memory and cognition [157]; *Lactobacillus reuteri* prevents children from developing autism when exposed to a high-fat diet in pregnant mice [158]; *Bacteroides fragilis* can restore intestinal permeability, change the composition of microbiota, and ease ASD symptoms [159]. *Bacillus coagulans* MTCC 5856 can improve depression and gastrointestinal symptoms in patients with severe IBS. *Bifidobacterium longum* NCC3001 shows similar effect in IBS patients [39].

With regard to cancer, studies indicate that the potential of microbes in the treatment of lung, ovarian and colon cancer could be investigated. *Enterococcus hirae* and *Barnesiella intestinihominis* increases the longer progression-free survival of chemotherapy [160]. *Lactobacillus casei* ATCC334 can produce iron pigment, which plays a role in inhibiting tumor progression by activating the JNK signaling pathway [161].

Intestinal microbiota also plays an important role in the treatment of hypertension, cardiovascular and other diseases. Bacteria can produce multilayered agglomerations, called biofilms, which protect them from the physical stress of fluid flow, epithelial turnover in intestine lumen and help spread resistance genes [162]. Biofilms can occur throughout the oral, intestinal tract and in the appendix [163]. The presence of mucosal biofilms in a healthy gut can enhance host defenses, contribute to synergy between hosts and bacteria and contribute to nutrient exchange [164].

The role of biofilms in disease-related microorganisms in IBD, CRC, intestinal damage, stomach infection and oral diseases has been confirmed [165,166]. When bacteria colonize at the wound and form biofilms, they can slow or prevent the wound from healing. Microscopic results showed that patients with IBD had dense *Bacteroides fragilis* biofilms [167]. IBD are also associated with an imbalance of microbiota and the destruction of mucosal epithelium, thus promoting species migration [168]. Biofilms help pathogens escape their hosts’ defenses, which can lead to disease progression [169]. Many anaerobic and aerobic microorganisms, such as *Klebsiella pneumoniae*, *E. coli,* and *Fusobacillus nuclei* settle in the intestines and cause wounding all over the body in IBD patients [170].

The biofilm on the mucosa of Familial Adenomatosis Polyopsis (FAP) patients acts as a marker of CRC [171]. Bacteria that form pathogenic biofilms can be used as signals to detect diseases. These can be categorized as (1) Oral markers: *Porphyromonas gingivalis, F. nucleatum*, *Prevotella*; (2) Stomach marker: *H. pylori*, (3) Intestinal markers: CRC: *F. nucleatum*, enterotoxigenic *Bacteroides fragilis* (ETBF, pks+ *E. Coli*, N1N12 diacetyl-spermine; Gut wounds: *F. nucleatum, E. Coli* and *K. pneumoniae*. *E. coli* were isolated from the intestines of colitis carrying mice with observed concurrent DNA damage and cancer [172]. *Clostridium jejuni* also damages the polarity of TLR9, which in turn destroys the epithelial barrier and increases CXCL8 production [173]. *Giardia* sp. and *C. jejuni* can advance the release of plankton microorganisms, and *C. jejuni* can activate the potential toxic genes of *E. coli* to promote their adhesion in human intestinal cells. This process is aided by the up-regulation of pro-inflammatory interleukin 8 (CXCL8) expression and down-regulation of TLR4 expression [173]. These findings may shed light on how intestinal pathogens can transform symbiotic bacteria into pathogens during the acute phase of infection.

### 3.2. Metabolites

#### 3.2.1. SCFAs

Anaerobic microorganisms ferment undigested food and other host metabolites in the large intestine to produce beneficial SCFAs [174,175]. SCFAs (acetate, propionate, and butyrate) are absorbed from the intestinal cavity, but their subsequent distribution and metabolism within host cells are different. Butyrate mainly provides energy for epithelial cells, propionate is mainly responsible for metabolism in the liver and acetate can exist in a higher concentration in the peripheral blood for long time [176]. Intracellular butyrate and propionate both restrain histone deacetylase (HDACs) activities in immune cells and promote histone hyperacetylation, thereby affecting gene expression and cell differentiation, such as the down-regulation of IL6 and IL12.

Recent studies have shown that SCFAs have some anti-inflammatory effects and can modulate regulatory T cells in the colon of mice [177]. Butyrate and propionate induce the differentiation of regulatory T cells (TReg) to activate the transcription factor FOXP3, which plays a key role in overcoming intestinal inflammation. SCFAs can inhibit histone deacetylase (HDAC) by acting on Ffar2 receptors on the surface of colon TReg cells, thus promoting the proliferation and function of Treg [177]. Low levels of butyric acid may boost CRC and may also alter gut microbiome diversity [178].

Extracellular SCFAs interact with host cell surface receptors. All host cells expressed GPR41 (FFA1), GPR43(FFA2) and GPR109A. GPR43 interacts with propionate, butyrate and acetate, GPR41 interacts with propionic acid in strongest saline condition [176] but GPR109A only interact with butyrate [179]. The interaction of butyrate with GPR109A induces an anti-inflammatory effect of butyrate by promoting the differentiation of TReg cells and T cells [180]. The interaction of acetate and propionate with GPR43 also leads to anti-inflammatory effects by regulating TReg cells [177]. GPR43 and GPR109A are tumor suppressor genes that induce the anticancer effect of propionate and butyrate derivatives. Butyrate can also play an anticancer role by restraining the proliferation of and selectively promoting apoptosis of CRC cells [62].

GPR43^−/−^ antigen-specific Th1 cells can cause more severe colitis in recipient mice. SCFAs activate STAT3 and mTOR pathways in Th1 through GPR43 and up-regulates the expression of transcription factor Blimp1, thus promoting Th1 to produce IL-10 and alleviating the colitis of mice [181]. SCFAs also promote the generation of IL-10 in T cells of patients with IBD and the oral administration of SCFAs in mice can alleviate colitis. SCFAs also induce Intestinal Epithelial Cells (IEC) RegIIIγ and β-defensin in a GPR43 dependent manner [181]. However, the intestinal microbiota can be reconstructed by treating potential pathogens and recovering butyrate producing bacteria [182]. Other molecules, such as many phenolic compounds have been reported to have antimicrobial effects and alter the diversity of intestinal microbiota [183]. In addition, the microbiomolecular patterns of some symbiotic bacteria (MAMPS) alter specific signaling pathways to increase inflammation.

#### 3.2.2. Small Molecules of the Microbial Enzymatic Pathways

Microbes in the gut can affect the health, but the proteases secreted by these microbes are also very important in developing diseases, such as arterial sclerosis, skin disease, enteritis and cardiovascular disease, etc. Intestinal bacteria converts dietary choline into trimethylamine (TMA) [184] which is subsequently converted to trimethylamine oxide (TMAO) by the metabolism in the liver. TMAO can induce cardiac hypertrophy and fibrosis in rats. This further increases the risk of atherosclerosis and, inhibits the mTOR pathway, exacerbating aging and cognitive impairment. A microbial TMA lyase inhibitor does not affect the viability of the symbiotic bacteria, but it can transform the CutC/D gene product into a potent inhibitor to prevent choline to TMA formation and TMAO generation from food. Simultaneously, it reduces platelet aggregation, thrombosis, and reverses the bacterial microbiota alteration induced by high choline food and increases nontoxic *A**. muciniphilla* bacteria.

*M. globosa* is a common skin color fungi, secrete proteinase MgSAP1, associated with the host physiology [185]. MgSAP1 rapidly hydrolyses *Staphylococcus* protein A (SpA) and prevents *S. aureus* biofilm formation, thereby inhibiting the bacteria and fungi from harboring in the skin, to help maintain a healthy skin. Xanthine Oxidase (XO) polysaccharide (EPS CG11), isolated from *Lactobacillus* BGCG11, prevents hyperalgesia, edema, and inhibits the inflammatory response. EPS CG11 regulates the expression of related inflammatory factors, such as the decreased expression of IL-1β, TNF-α, iNOS and increased anti-inflammatory IL-10 without affecting neutrophil infiltration [186]. Thus, EPS CG11 can be used to develop new analgesic drugs. AimA is an immuno-regulatory protein that is secreted by *Aeromonas* [187]. These symbiotic bacteria inhibit harmful bacteria and intestinal inflammation in the host. In a zebrafish model, this bacterium reduces inflammation and in a chemical model, AimA prevents the excessive buildup of neutrophils and septic shock.

Some bacteria also secrete amino acid-derived antibiotics to fight diseases. Gut Bacteria, *Clostridium scindens* and *C. sordellii* secrete tryptophan derived antibiotics 1-Acetyl-β-carotine and turbomycin A respectively [188]. These two antibiotics inhibit the growth of *C. difficile* and other intestinal bacteria by inhibiting the formation of the diaphragm during bacteria’s split phase. *Olsenella scatoligenes* can secrete indoleacetic decarboxylase in the intestinal tract, which plays a key role in the process of tryptophan fermentation to form faecosine [189]. Intestinal tract Actinobacterium *Eggerthella lenta* possess glycoside reductase operon that can inactivate its natural toxins digoxin [190].

#### 3.2.3. Toxic Metabolites

The bacterial fermentation of aromatic amino acids produces a range of metabolites, some of which are toxic, including certain nitrogen compounds, ammonia, amine and sulphides [191]. Some nitrogenous compounds, especially nitrites, increase the risk of cancer through DNA alkylation [192]. Ammonia is also a carcinogen at low concentrations and has been shown to be associated with mucosal damage and colorectal adenocarcinoma in animal models [193]. A high concentration of polyamine is toxic, and is involved in various diseases such as oxidative stress and cancer [194]. Pathogens, such as *Shigella flexneri*, *Priceenterica* subsp., *Streptococcus pneumoniae*, *H. pylori* and *Salmonella enterica serovar Typhimurium* employ polyamines to enhance their virulence [195]. A small amount of sulfate reducing bacteria (such as *Desulfovibrio*) uses lactate salts as a common substrate for growth and forms sulfide [196]. Sulfide is not only toxic to colon cells but also inhibits the oxidation of butyrate and thereby destroys the integrity of the colon cell barrier. At a very low concentration (0.25–2 mM), sulfide produces ROS that is known to induce DNA damage, resulting in cancer development [197].

#### 3.2.4. Bile Acid Metabolism

A high-fat diet is associated with an increased incidence of CRC with increased bile secretion and increased stool bile acid concentration [198]. Bile acids have strong antibacterial activity and may affect the composition of intestinal microbiota. The primary bile acid transforms into several different secondary bile acids, mainly deoxycholic acid and lithocholic acid [199]. Deoxycholic acid can promote liver cancer [200]. Mice, when kept on a diet with deoxycholic acid, showed a reduced generation of SCFAs and an altered microbiota diversity, such as the reduction of bacteroides, an increase of *Gammaproteobacteria* and certain firmicutes [201]. *Clostridium* species in the intestinal symbiotic bacteria may inhibit the immune response of the host at the liver by metabolizing the primary bile acids into secondary bile acids. A high-fat diet increases intestinal *Clostridium* bacteria and secondary bile acids in the liver that lead to promote liver cancer [202]. *Fusobacterium* increases the primary bile acid in hepatic sinus endothelial cells expressing more CXCL16. CXCL16 causes the accumulation of the CXCR6^+^ NKT cells in the liver, the production of cytokines (such as IFNɤ), and inhibition of liver tumor. NKT cells in the human liver activate NK+/in Mucosal Associated Invariant T cells (MAIT) for its anti-cancer effect [203]. Secondary bile acids are more capable of destroying cell membranes and, destroy membrane-related proteins (NADPH oxidases and phospholipase A2) that cause ROS production [204]. For example, bile acids interact with nuclear receptors to activate signaling pathways that induce apoptosis. However, some bile acids also as assist with detoxification. Ursodeoxycholic acid seems to inhibit ROS and protects cells from deoxycholic acid. Regulating bile acid production could be a way to improve cancer treatment.

## 4. Treatment and Adjustment

An imbalance in intestinal microbiota affects human health. Therefore, restoring the balance of the intestinal environment could be a potential therapeutic target. Recent studies have shown that FMT can effectively achieve microbiota reconstruction, which plays a role in prevention, mitigation and treatment for health. In addition, prebiotics, antibiotics and small molecules of the microbial enzymatic pathways can also be used to regulate intestinal microbiota to improve disease symptoms.

### 4.1. Fecal Microbial Transplantation (FMT) and Oral Administration

Fecal Microbiota Transplantation (FMT) is a method to treat gastrointestinal diseases by transplanting the fecal filtrate of healthy donors to the gastrointestinal tract of the recipients to restore the diversity of microbiota [205]. At present, thousands of cases of *C. difficile* infections (CDI) [206] and IBD have been successfully treated [207]. For patients with recurrent *C. difficile* infection, FMT can prevent recurrence, and their urethral infections are subsequently relieved or cured [205,208]. In a similar case, a 19-year-old woman with Crohn’s disease with *C. difficile* infection was relieved after two FMTs from the mother [209]. When two alopecia patients were treated with FMT, their *C. difficile* infection was cured and hair regrowth occurred, suggesting that intestinal microbiota plays an immunomodulatory role in these autoimmune symptoms [210]. Similarly, patients with chronic colitis with severe abnormal distension, who were uncured by antibiotics, found their health to have improved after FMT treatment [211]. Other results also indicated that the cure rate of ulcerative colitis (UC) treated with antibiotics was significantly improved after FMT [212]. The treatment of refractory type II celiac disease is very difficult. It is worth noting that when treated with FMT, symptoms of celiac disease disappeared completely and the duodenal villi were completely recovered [213]. In recent years, the incredible therapeutic effects of microorganisms on neurological, cancer and metabolic diseases have been widely reported. In neurological diseases, FMT increased the diversity of bacterial microbiota and the number of beneficial bacteria in patients with hepatic encephalopathy and improved their cognitive ability [214]. *Bifidobacterium adolescentis* can treat constipation [215]. Adult patients with Slow Transit Constipation (STC) had a 30 percent increase in their clinical cure rate without any adverse reactions after FMT compared to conventional treatment.

By sequencing its genome and through further molecular studies, we observed that bifidobacteria had an anti-tumor effect. They presented a similar effect to anti-PD-L1 immunosuppressive inhibitors when mice were orally treated with bifidobacteria, and the combined treatment almost completely inhibited tumor growth [216]. It is worth noting that *A. muciniphila*, a new functional microbe with probiotic characteristics, has been found to be a potent inhibitor of a variety of diseases [217,218,219,220]. In humans and mice, the abundance of *A. muciniphila* in the intestinal microbiota was negatively correlated with T1D, suggesting that this bacteria had a protective effect in T1D [221]. Oral *A. muciniphila* bacteria can improve the liver function of streptozotocin-induced diabetic rats and restore normal intestinal microbiota, thereby relieving T2D [88]. *A. muciniphila* can heal liver injury in alcoholic liver disease patients and prevent obesity and its complications [222]. It can also improve the anti-tumor effect on lung/kidney cancer in mice when combined with PD-1 inhibitors [223]. Interestingly, the deactivated *A. muciniphila* or the pasteurized AMUC-1100 protein isolated from this bacteria’s outer membrane still had partial probiotic function and improved the metabolism of mice [224].

A recent study found that when staphylococcal nuclease (SNase), expressed by recombinant *Lactococcus lactis*, was orally administered in NOD mice, it delayed diabetes, and reduced the rate of morbidity and mortality [225]. SNase can be effective for islets of Langerhans and helps to reduce inflammation in the small intestine of NOD mice. Oral administration of *L. rhamnosus* JB-1 relieves stress induced anxiety and reduces stress related dendritic cell activation [226]. Oral administration of *E. Coli Nissle* 1917 (ECN) can reduce MOG_35–55_ (MEVGWYRSPFSRVVHLYR NGK) peptides in induced Experimental Autoimmune Encephalomyelitis (EAE) [227]. The alleviation of the disease may be associated with a decreased secretion of autoreactive CD4 T cells inflammatory factors in peripheral lymph nodes and in the central nervous system.

Oral probiotics are shown to continuously increase the number of *Lactobacillu*s and bifidobacteria in fecal microbiota of premature infants with a low birth weight [228]. In addition, oral a *Lactobacillus/*lactoferrin mixture can treat vaginal microbiota disorders [229]. Nasal inoculation of *L. rhamnosus* GG can effectively prevent allergic asthma caused by birch pollen in mice [230].

### 4.2. Dietary Interventions

#### 4.2.1. Dietary Fiber

Diet and nutrition greatly affect intestinal microbiota responsible for ferment dietary fiber to produce SCFAs, which have a variety of health benefits. Soluble and insoluble dietary fiber have a variety of functions, such as reducing bowel passage time, maintaining normal blood cholesterol levels, reducing the risk of coronary heart disease, colorectal cancer, postnatal blood glucose response and preventing harmful colonization [231]. Long-term dietary fiber consumption can reduce the risk of fecal inconsistencies in elderly women and significantly improve blood glucose level in patients with gestational diabetes [232]. Non-degradable fibers inhibit autoimmune diseases of the central nervous system by regulating the gut microbes [233]. A Mediterranean diet can reduce a woman’s risk of stroke and adjust the composition of mammary gland microbiota by improving its metabolites compared to a western diet [234]. A whole grain diet reduces inflammation, body weight, and has no effect on gut microbiota and its metabolites, thus increasing insulin sensitivity [235].

Dietary soluble corn fiber was significantly correlated with increased calcium absorption in adolescent females and the diversity of intestinal microbiota. The proportion of *Cymbidium* consumption with the edible soluble corn fiber increases the calcium absorption alongside an increase of *Clostridium* [236].

Dietary components can control the colonization of intestinal symbiotic bacteria. Seaweeds are a good source of nutrients such as proteins, vitamins, minerals, and dietary fibers [237]. Seaweed can affect the colonization of the intestinal tract of *Bacteroides plebeius* DSM17135 [238]. Dietary fiber supplementation, especially fructan and oligogalactose, increases the abundance of bifidobacteria and *Lactobacillus* in healthy adult feces, but does not affect its diversity [239].

#### 4.2.2. Probiotics and Prebiotics Products

The concept of prebiotics has been proposed for more than 21 years [240]. Preclinical data have shown that it can recover the intestinal tract damage, metabolism abnormalities, reduce the incidence of colorectal cancer, immune function and glucose intolerance. Prenatal supplements of symbiotic bacteria reduce the serum insulin concentration of pregnant women and reduces the risk of pre-eclampsia and dyslipidemia [241]. Probiotics reduce allergy in infants but are not effective for necrotizing enterocolitis (NEC) in premature infants.

The supplementation of the biogenetic elements, short chain oligogalactose (scGOS), long chain oligofructose (lcFOS) and m-16v bifidobacteria, increased the proportion of bifidobacteria for caesarean babies, but decreased the proportion of *Enterobacteriaceae* and fecal pH [242]. Long-term consumption of probiotics can improve cardiac dysfunction in obese rats [243]. Adding corn bran or wheat bran can improve the growth of weaned pigs by changing their intestinal microbiota and promoting the production of butyric acid [244]. In some cases, dietary adjustment alone cannot improve the disease, and probiotics with dietary intervention show better treatment option and prevention of the disease [245]. The gluten-free and casein-free diet have a limited effect on improving intestinal symptoms in children with autism. A restricted diet (no gluten and casein) with prebiotics (immuno-marker clasado^®^) Galactosyl Oligosaccharides (B-GOS^®^) jointly intervene and can alleviate autism [246]. Fructo-oligosaccharides (FOS), a dietary fiber found in fruits and vegetables, improved blood glucose homeostasis in animal models. The addition of Oligoxylose in rice was found to improve the balance of human microbiota [247]. Dietary supplements of mixed prebiotics have been shown to increase the bioavailability of heme iron without affecting the bioavailability of non-heme iron [248]. The combination of probiotics and oligosaccharides can improve the composition of colon bacteria and the health of the host. Inulin is a type of polyfructose that can be used for prebiotics, as a sugar substitute, emulsifying agent and fat substitute. In mice fed with a high-fat diet, inulin rebuilds damaged gut microbiota, restores the expression of antimicrobial genes, produces IL22 and proliferates intestinal epithelial cells [249].

The carbohydrates available to bacteria in dietary fiber, such as Microbiota-Accessible Carbohydrates (MAC diet) are important for increasing the diversity of intestinal microbiota. A low MAC diet resulted in the decreased diversity of intestinal microbiota in mice and, whereas it was found that a the high MAC diet could restore the diversity of microbiome [250]. Food containing complex carbohydrates or inulin in the MAC diet are the only sources of carbohydrate that can inhibit *C. difficile* infection in mice (CDI) [251]. The combination of inulin, oligosaccharides and the active role of *Lactobacillus* restored the composition of intestinal microbiota in mice with leukemia and also reduced the proliferation of liver cancer cells, muscle atrophy and improved their survival [252]. A randomized clinical trial showed that a combination of probiotic supplements containing five strains of *Lactobacillus* and *Bifidobacterium* could improve the symptoms of celiac disease in patients with IBS [253]. It was also observed that the patients had increased intestinal *Lactobacillus*, *Staphylococcus* and *Bifidobacterium*, which related to the changes of intestinal microbiota. The combined therapy of FMT and oral pectin administration can significantly improve the condition of slow transit constipation [254].

### 4.3. Drug Adjustment

#### Drugs Have Therapeutic Effects on Some Diseases Caused by Microbial Disorders

Metformin is a commonly used drug for metabolic disorders [255]. Metformin increases the abundance of slime producing bacteria (*Akkermansia*), SCFAs producing bacteria and decreases the abundance of *Clostridium nucleatum* and Firmicutes. Metformin regulates bacteria to increase insulin sensitivity, and the GLP-1 released by L cells, affects mitochondrial complex I and activates duodenal AMPK. It also inhibits liver heterogeneity and increases GLP-1 levels, thereby reducing blood glucose. It can also delay aging by enhancing the intestinal barrier function, increasing the level of SCFAs, reducing pro-inflammatory bile acids and inhibiting inflammation. Microbial metabolites are also effective in suppressing cancer and for treating intestinal inflammation. Food rich in histidine can enhance the activity of low dose methotrexate in mice, reveals a tumor inhibitory effect and reduces drug toxicity. The addition of histidine in food may provide a feasible method through which to improve methotrexate treatment [256].

Antibiotics usage, that relieves intestinal inflammation, has been widely reported on. If the antibiotic treatment time is transitory, the bacteria develop resistance. *Enterobacteriaceae* infections show resistance within 48 h of treatment of β-lactams antibiotic (APBL) therapy [257]. Again, it cannot target specific harmful bacteria. Antibiotic use was found to significantly reduce overall survival in patients receiving checkpoint inhibitors for advanced cancer. A significant finding is that antibiotics impart more harmful than beneficial effects in the recovery of intestinal microbiota and for curing diseases.

## 5. Conclusions

The microbiome varies in person to person, exists in the whole life cycle of an individual, and is affected by the environment, diet, lifestyle and other factors. In recent years, numerous studies have demonstrated that gut microbes greatly contribute to maintaining human health and cure diseases. The targeted regulation of intestinal microbiota is expected to become a powerful weapon to treat diseases that are caused by inflammation. Probiotics, FMT, oral pills, aerosol therapy and other methods can be used to intervene intestinal microorganisms to achieve the objectives of treatment. In particular, the dietary probiotics can increase the abundance of beneficial bacteria in the intestinal tract, thus becoming an effective treatment for intestinal diseases. In addition, the combination treatment of certain nutrients with probiotics also contributes to the reconstruction of intestinal microbiota.

However, there are still some potential risk factors. The sourcing of bacteria in FMT treatment may be unstable, which questions whether probiotics have any adverse reactions in the human body. Recently, two major articles highlighted that probiotics damage the reconstruction of intestinal mucosal microbiome when treated with antibiotics. As a result, the intestinal mucous membrane often develops resistance to probiotic bacteria’s colonization [253,258]. This illustrates that the clinical application of probiotics also needs to be implanted with caution. Again, probiotics for different subjects should be tested with various conditions to develop a more comprehensive understanding of their effects. Host-microbe interaction should be studied by intestinal mucosa sampling to develop a better prediction model.

In the gut, host-microbe interactions utilize microbes in a way that is beneficial. However, further research could open avenues for the further treatment of human diseases in the future. Understanding the composition of microbial groups, filtering harmful fungi or bacteria, exploring the development of the disease or reducing the disease symptoms can assist disease improvement by adjusting new therapy with microbiota. Therefore, an in-depth exploration of beneficial gut microbes and their functional patterns prove useful in combatting more complex diseases.

In future, microorganism signatures can be explored and used as biomarkers for specific diseases. It will be also possible to use intestinal microbes for the diagnosis, prevention and treatment of diseases. In this review, we presented useful bacteria, more specifically, *A. muciniphila,* as an immensely beneficial bacterium (Figure 4). It is associated with both metabolic and intestinal diseases and plays an effective role in obesity and diabetes. Based on the beneficial effect of *Akkermansia* on human health, a company has been established (A-mansia biotech, founded by Prof. Willem M. De Vos) to provide pasteurized bacterial supplements to improve diabetes and obesity. The study of *A. muciniphila* and the mechanism of its function are expected to provide a foundation for breakthrough treatment in several diseases.

## Figures and Tables

**Figure 1 ijms-22-12661-f001:**
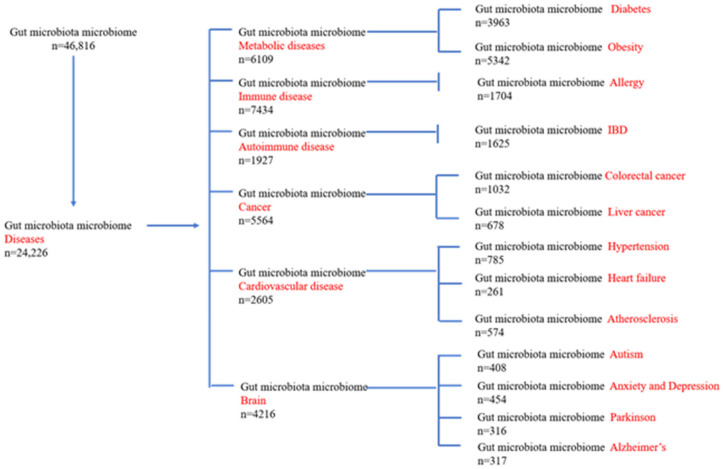
A PRISMA flow diagram of articles in PUBMED for gut microbiota. Texts in the diagram are used in search item and number (n) of articles showed up in PUBMED.

**Figure 2 ijms-22-12661-f002:**
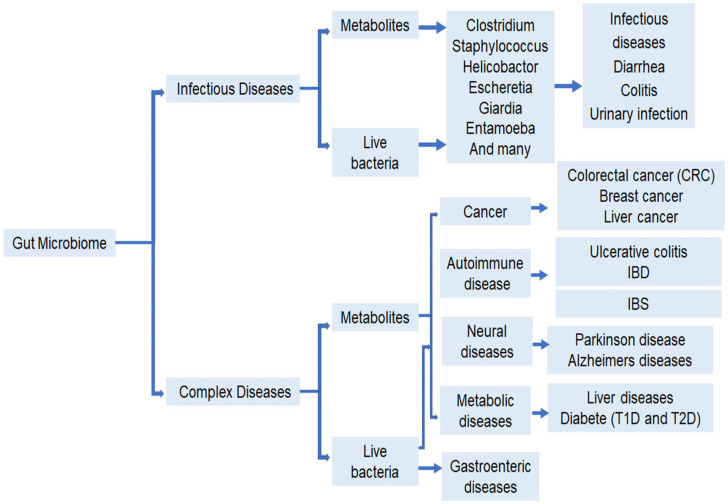
Gut microbiome in health and diseases. Gut microbes have impacts on both infectious diseases and complex diseases. Their metabolites can be directly involved to induce disease symptoms, for examples, diarrhea, colitis, etc. Although the gut microbiome may not directly induce complex diseases, such as CRC, autoimmune diseases and neuronal diseases, etc., they have profound effects on disease symptoms. As complex diseases occur due to interaction of genetic and environmental factors, gut microbes and their metabolites could play important roles as environmental factors and to help in shaping or reducing these symptoms. However, antibiotics affect the microbial population and have an overall adverse effect on human health.

**Figure 3 ijms-22-12661-f003:**
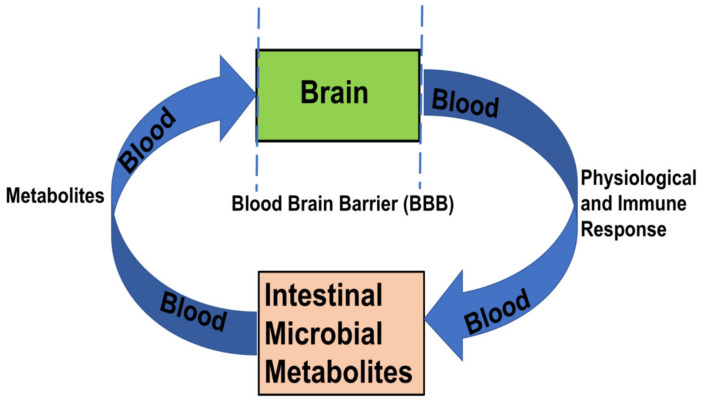
**Feedback mechanism of axis of gut-brain.** Metabolites secreted by microbiotas are absorbed in gut epithelia, through which they can reach the brain through bloodstream by crossing BBB. They activate the brain to secret hormones and neurotransmitters, which can regulate gut microbes and intestinal activities.

**Figure 4 ijms-22-12661-f004:**
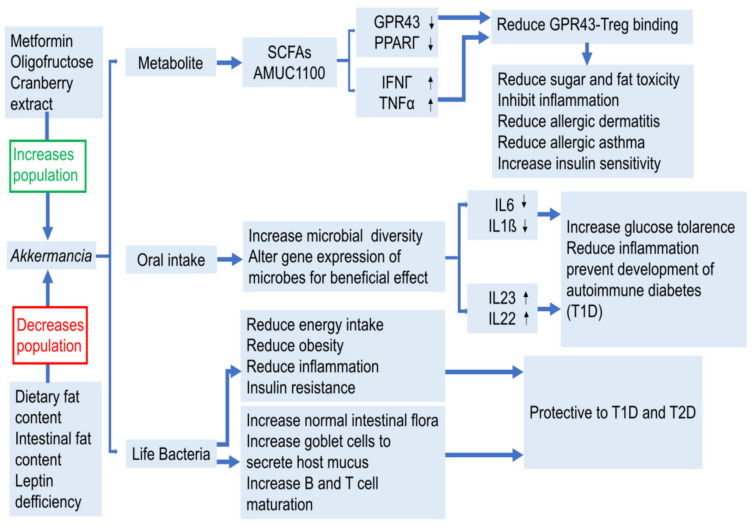
The role of *A. muciniphila* in regulating various diseases. *Akkermansia* appears to be a beneficial bacterium for human health and is associated with many disease symptoms. Its population increases through dietary constituents such as cranberry extract or drug metformin whereas decreases as a result of a high fat diet or intestinal fat content. *Akkermansia* metabolites increases SCFAs that in turn regulate GPR43 or IFNγ to inhibit Treg, which reduces inflammation, symptoms of asthma and allergy. They also help to enhance metformin effect to increase insulin sensitivity. It also plays a role to reduce inflammation and development of T1D. FMT of *Akkermansia* has been proved to prevent or delay the onset of T1D and T2D. *Akkermansia* transplantation could be developed as a potential therapeutic agent for diabetes, allergy and asthma.

## Data Availability

All data will be available upon publication.

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
