# Peer review of "Implications of Gut Microbiota in Complex Human Diseases"

_ijms, 2021, doi:10.3390/ijms222312661_

Round 1

Reviewer 1 Report

The manuscript is further improved and has several features that I find attractive. These include figure 1 and table 1.
The manuscript is still very light on the discussion. Its descriptive nature will not attract readers who are new in the field. Nevertheless, it may provide more experienced readers with useful information.
Oversimplifications/superficial discussion is still present. Although it does not constitute a major problem for experienced readers (who can filter them out but), it creates a risk for people who are being introduced to this field to overinterpret certain data. Things are not as black&white as presented here, and caution should be taken when describing them (even with a simple choice of words, such as “might be” instead of “is”).
The manuscript needs extensive English editing. Many sentences are grammatically incorrect, or suboptimal. These include “highlights” – point 1, line 41, line 52, just to name a few.
Finally, it is not appropriate to include a statement of the conflict of interest in the main text – this reads more like an advertisement. This should be stated in the section “conflict of interest”.

Author Response

The manuscript is further improved and has several features that I find attractive. These include figure 1 and table 1.
The manuscript is still very light on the discussion. Its descriptive nature will not attract readers who are new in the field. Nevertheless, it may provide more experienced readers with useful information.
Oversimplifications/superficial discussion is still present. Although it does not constitute a major problem for experienced readers (who can filter them out but), it creates a risk for people who are being introduced to this field to overinterpret certain data. Things are not as black&white as presented here, and caution should be taken when describing them (even with a simple choice of words, such as “might be” instead of “is”).

Ans. Thanks for your comments. We extensively edited the manuscript by removing sentences that are descriptive.

Especially conclusions are less weighted by removing “is” to “may be” or “might be”.

The manuscript needs extensive English editing. Many sentences are grammatically incorrect, or suboptimal. These include “highlights” – point 1, line 41, line 52, just to name a few.

Ans. The manuscript now extensively edited and grammatical corrections are introduced. Suboptimal sentences are corrected. Other spelling and grammatical mistakes are also corrected in revised manuscript.

Finally, it is not appropriate to include a statement of the conflict of interest in the main text – this reads more like an advertisement. This should be stated in the section “conflict of interest”.

Ans. Thanks. Conflict of interest has been removed from the main text.

Reviewer 2 Report

I have no additional comments

Author Response

I have no additional comments

Ans. Thank you very much.

Reviewer 3 Report

I enclose my comments as pdf file.

Author Response

General comments

The article was much improved. The title now sounds better. Due to the large number of issues raised, it is briefly written, but can be treated as a general overview, helpful for many authors starting in the topic. Given the above, I would not exclude the possibility of publishing this article in the Journal.

Detailed comments

  • Lines 68, 70: give a dot after “spp”; in lines 238, 436, 438 “spp.” it should be written without italics; in lines 519, 520 (2x) and 520 (2x) it should be written with a small letter.

Ans. Thanks for your comments. We corrected them in the revised manuscript.

  • Line 696 “subsp” - give a dot.

Ans. We corrected it.

  • Some parts of the manuscript have a different font than required. It needs to be corrected.

Ans. Corrected.

  • Figure 2 caption: what does it mean “in most cases” – on the basis of what is the statement? It would be more scientific, if Authors gave some references here. It must be corrected.

Ans. Thanks for this comment. We removed “in most cases”

  • Lines 556-557: give true/the latest definition of ‘probiotics’ with a reference. Definition given by authors is wrong. Maybe these links will be helpful: https://isappscience.org/for-scientists/resources/probiotics/

https://www.nature.com/articles/nrgastro.2014.66

Ans. Thanks for this comment. We introduced the current definition of “probiotics” by WHO in 2013 in the revised manuscript with reference.